# Distribution of bacteria and antimicrobial resistance in retail Nile tilapia (*Oreochromis* spp.) as potential sources of foodborne illness

Jarukorn Sripradite[1], Varangkana Thaotumpitak[2,3], Edward R. Atwill[4], Woranich Hinthong[5], Saharuetai Jeamsripong[2,3]*

**1** Department of Social and Applied Science, College of Industrial Technology, King Mongkut's University of Technology North Bangkok, Bangkok, Thailand, **2** Department of Veterinary Public Health, Faculty of Veterinary Science, Chulalongkorn University, Bangkok, Thailand, **3** Research Unit in Microbial Food Safety and Antimicrobial Resistance, Department of Veterinary Public Health, Faculty of Veterinary Science, Chulalongkorn University, Bangkok, Thailand, **4** Department of Population Health and Reproduction, School of Veterinary Medicine, University of California, Davis, Davis, California, United States of America, **5** Princess Srisavangavadhana College of Medicine, Chulabhorn Royal Academy, Bangkok, Thailand

* saharuetai.j@gmail.com

**Data Availability Statement:** All raw data are provided at the following DOI: 10.6084/m9. figshare.24532972.

**Funding:** This study was supported by Thailand Science Research and Innovation Fund

## Abstract

This study aimed to investigate AMR profiles of *Aeromonas hydrophila*, *Salmonella* spp., and *Vibrio cholerae* isolated from Nile tilapia (*Oreochromis* spp.) ($n = 276$) purchased from fresh markets and supermarkets in Bangkok, Thailand. A sample of tilapia was divided into three parts: fish intestine ($n = 276$), fish meat ($n = 276$), and liver and kidney ($n = 276$). The occurrence of *A. hydrophila*, *Salmonella*, and *V. cholerae* was 3.1%, 7.4%, and 8.5%, respectively. A high prevalence of these pathogenic bacteria was observed in fresh market tilapia compared to those from supermarkets ($p < 0.05$). The predominant *Salmonella* serovars were Paratyphi B (6.4%), followed by Escanaba (5.7%), and Saintpaul (5.7%). All isolates tested positive for the virulence genes of *A. hydrophila* (*aero* and *hly*), *Salmonella* (*invA*), and *V. cholerae* (*hlyA*). *A. hydrophila* (65.4%), *Salmonella* (31.2%), and *V. cholerae* (2.9%) showed multidrug resistant isolates. All *A. hydrophila* isolates ($n = 26$) exhibited resistant to ampicillin (100.0%) and florfenicol (100.0%), and often carried *sul1* (53.8%) and *tetA* (50.0%). *Salmonella* isolates were primarily resistant to ampicillin (36.9%), with a high incidence of *bla*$_{TEM}$ (26.2%) and *qnrS* (25.5%). For *V. cholerae* isolates, resistance was observed against ampicillin (48.6%), and they commonly carried *qnrS* (24.3%) and *tetA* (22.9%). To identify mutations in the quinolone resistance determining regions (QRDRs), a single C248A point mutation of C248A (Ser-83-Tyr) in the *gyr*A region was identified in six out of seven isolates of *Salmonella* isolates. This study highlighted the presence of antimicrobial-resistant pathogenic bacteria in Nile tilapia at a selling point. It is important to rigorously implement strategies for AMR control and prevention.

Chulalongkorn University (HEA663100108), National Research Council of Thailand (NTCT) (N42A660897), University of California Davis (A22-3754-S001), and Rachadapisek Sompote Fund Chulalongkorn University (GR_62_37_31_02) and College of Industrial Technology, King Mongkut's University of Technology North Bangkok (Res-CIT0232/2019). The funders had no role in study design, data collection and analysis, decision to publish, or preparation of the manuscript.

**Competing interests:** The authors have declared that no competing interests exist.

## Introduction

Fish consumption has been increased due to its perceived health benefits and affordability. However, it is important to note that microbial contaminants in fish have been identified as sources of foodborne outbreaks on a global scale. Additionally, the emergence of antimicrobial resistance (AMR) in aquatic products poses a novel human health risk, with improper storage and handling practices between fresh and supermarkets playing a significant role in this concern. Many AMR pathogens identified in aquatic animals are associated with human clinical incidents and environmental strains, highlighting the relevance of the One Health concept. In contrast to terrestrial animals, the surveillance and tracking of antibiotic resistant bacteria in aquatic animals is often overlooked due to the diverse range of aquatic species, cultivation methods, and sample sources. One of several attempts to mitigate AMR issues in aquaculture is the development of the integration of AMR surveillance in fish and aquatic products into the National Action Plan (NAP). This approach is particularly emphasized in numerous Asian countries [1]. Specific bacterial pathogens were identified as priorities for targeted surveillance, including notably *Aeromonas hydrophila*, *Salmonella* spp., and *Vibrio* spp., within the context of national AMR monitoring efforts in the aquaculture sector [2].

*A. hydrophila* is a frequently pathogenic bacterium in freshwater fish and is also associated with foodborne illnesses in humans. The occurrence of *A. hydrophila* in freshwater fish and aquatic products in retail settings has been relatively rare, as documented in countries such as Turkey (5.7%) [3]. Conversely, the prevalence of *A. hydrophila* has been extensively documented in diseased tilapia in Egypt, with reported rates of 53.4% [4]. Although the prevalence of *A. hydrophila* may be low, the emergence of multidrug resistance (MDR) and the presence of various virulence genes have been extensively reported in populations of clinically asymptomatic tilapia [5]. Integrons play a crucial role in the formation of *Aeromonas* MDR strains, as they enable the bacteria to carry different resistance genes within the same cassettes. The prevalence of MDR *Aeromonas* strains that carry integrons has been investigated in farmed tilapia in India [6]. The presence of integrons in conjugative plasmids can play a key role in facilitating the transmission of AMR within a bacterial community among animals, humans, and the environment.

*Salmonella* is not a native pathogen of fish and is not commonly found in fish. However, this pathogen can be introduced to tilapia during the pre-harvest stage due to water and environmental contamination. Similarly, post-harvest stages, including fish processing, transportation, and distribution, can also contribute to the introduction of *Salmonella* or other pathogenic bacteria into tilapia. Although not being a natural fish pathogen, *Salmonella* has been identified as a prominent cause of foodborne outbreaks linked to the consumption of fish. However, *Salmonella* has been the leading cause of foodborne outbreaks that were associated with fish consumption. For example, *Salmonella* contamination in Nile tilapia has been documented in various countries, with a prevalence up to 45.5% reported in Brazil [7]. Furthermore, these *Salmonella* isolates showed high resistance to antimicrobials that are typically prescribed for treating bacterial infections in hospital settings, such as erythromycin, tetracycline, and ampicillin [8]. Moreover, certain strains exhibited less effectiveness against the last line of antimicrobials, including colistin and vancomycin.

*Vibrio cholerae* are common microbiota in freshwater fish and the aquatic environment. It is hypothesized that *V. cholerae* colonized as normal flora in the gastrointestinal tract of fish and exists outside the host by adhering to phytoplankton or growing as free-living organisms. The low occurrence of *V. cholerae* in fish has been observed in Malaysia [8]. However, a previous study indicated that tilapia could potentially serve as a suitable host for *V. cholerae* resulting in posing a risk to human health [9]. In the past, *Vibrio* strains were typically susceptible to

commonly used clinical antimicrobials, including cephalexin and norfloxacin. However, a recent publication highlighted that *V. cholerae* residing in cultivated fish and natural water sources has developed more resistance to widely used antimicrobials in human medicine such as amoxicillin, ampicillin, erythromycin, and co-resistance to other potent antimicrobials such as aminoglycosides, carbapenems, and quinolones [10]. This situation presents a significant public health concern [10, 11].

Extended-spectrum β-lactamases (ESBLs) are enzymes produced by bacteria that can hydrolyze a wide range of beta-lactam antibiotics, including high-generation cephalosporins and aztreonam, while also showing co-resistance to various classes of antimicrobials. This is particularly concerning as it leads to a restricted array of effective antimicrobial treatments, which may result in treatment failures. Additionally, it is important to regularly investigate mutations occurring in quinolone resistance-determining regions (QRDRs) due to their significant impact on the development of quinolone resistance. In Thailand, quinolones, including enrofloxacin, oxolinic acid, and sarafloxacin, are approved for the treatment of prevalent bacterial illnesses in fish, such as columnaris, motile aeromonas septicemia, and vibriosis [12]. However, *Salmonella* exhibited high resistance to quinolones. For example, three-quarters and one-thirds of *Salmonella* isolated from a tilapia farm resisted oxolinic acid and enrofloxacin, respectively [13]. DNA gyrase and topoisomerase IV, which comprise four essential subunits, including *gyr*A, *gyr*B, *par*C, and *par*E, are the primary targets of quinolones, acting to inhibit bacterial replication. The structural change at these sites can lead to a resistance profile in the decrease of quinolones. The major mechanism conferring resistance is chromosomal mutations in the *gyr*A and *par*C subunits. A frequent mutation that occurs at position 83 (Ser to Ile) in *gyr*A is particularly common and leads to a high level of resistance to quinolones. On the contrary, other amino acid substitutions confer lower levels of resistance [14]. Examining these mutations is essential because they directly impact the reduction of susceptibility to quinolones. In addition, plasmid-mediated quinolone resistance (PMQR) genes can confer a lower level of quinolone resistance, which is primarily responsible for the horizontal transfer of quinolone resistance among bacterial populations.

Microbial diversity in Nile tilapia exhibited variations depending on its ecological niche within the fish's organs. *Salmonella* spp. and *Vibrio* spp. were the predominant bacterial species identified in the intestinal tract of fish. However, *A. hydrophila* was the most prevalent bacteria in the kidney, liver, and spleen of diseased fish [15]. Therefore, it is recommended to collect samples from various sections of fish organs for AMR monitoring and surveillance, given the diversity of bacterial populations in fish. Virulence genes (*A. hydrophila*: *aero* and *hly*; *Salmonella*: *invA*; *V. cholerae*: *tcpA*, *ctx*, and *hlyA*), were chosen based on their prevalence and linked to the pathogenesis of diseases in humans. Few studies have conducted investigations into AMR pathogens among retail tilapia intended for human consumption, and the risk of AMR transmission from tilapia consumption has been insufficiently documented in Thailand [16]. Therefore, this study aimed to examine the prevalence of resistance phenotype, genotype, and virulence genes in *A. hydrophila*, *Salmonella* spp., and *V. cholerae* obtained from Nile tilapia acquired from both fresh markets and supermarkets in Bangkok, Thailand.

## Materials and methods

### Sample collection and preparation

A total of Nile tilapia (*Oreochromis* spp.) ($n$ = 276) were collected from fresh markets ($n$ = 151) and supermarkets ($n$ = 125) from October 2019 to November 2020. The Nile tilapia specimens were purchased from five districts within Bangkok, consisting of Pom Prap Sattru Phai, Samphanthawong, Din Daeng, Thonburi, and Klongsan. These sampling sites were chosen because

of their significant density of human population. One to two fish were collected from each establishment, and the storage conditions at the point of sale (including the presence of ice, ambient air temperature, and relative humidity) were recorded.

Each individual Nile tilapia was carefully packed within two layers of sterile plastic bags, ensuring a double layer of protection. Subsequently, they were stored within a refrigerated container along with crushed ice to maintain a temperature range of 4–8˚C during transportation. All the samples were submitted and processed within 6 hr after collection in the Department of Veterinary Public Health, Faculty of Veterinary Science, Chulalongkorn University.

Each tilapia was weighed and then dissected aseptically to collect three different types of samples, including fish meat ($n$ = 276) fish intestine ($n$ = 276), and liver and kidney tissues ($n$ = 276) for further microbiological examination.

## Isolation and confirmation of *A. hydrophila*

The isolation of *A. hydrophila* was carried out according to the guideline of the Department of Public Health of England with slight modifications [17]. A swab of each fish sample was directly streaked onto Rimler-Shotts (RS) medium base (HiMedia Laboratories Ltd., Mumbai, India) supplemented with novobiocin at a concentration of 5 mg/l. The plates were incubated at 37˚C for 24 hr. The typical colony of *A. hydrophila* appears in a light yellow and round shape on the plate. In each positive sample, three presumptive colonies of *A. hydrophila* were verified using biochemical tests, including triple iron sugar (TSI), indole production, and oxidase tests (Difco, MD, USA). Each *A. hydrophila* isolate was individually purified using tryptic soy agar (TSA) (Difco). One bacterial isolate per positive sample was selected for the antimicrobial susceptibility test (AST).

## Isolation and serotyping of *Salmonella*

The method for *Salmonella* isolation was followed by ISO 6579–1:2017 [18]. Briefly, samples of fish meat (25 g) intestine (1 g), and liver and kidney tissues (1 g) were weighed and packed separately. For tilapia meat samples, 225 ml of buffered peptone water (BPW) (Difco) was added, while the intestinal tract, and liver and kidney were added to 9 ml of BPW (Difco). The samples were homogenized using a stomacher (Stomacher®400 Circulator, West Sussex, UK) and incubated at 37˚C for 24 hr. Approximately, 0.1 ml of the suspension from BPW was inoculated in modified semisolid Rappaport-Vassiliadis (MSRV) (Difco) plates and MSRV-positive plates were restreaked onto xylose lysine deoxycholate (XLD) (Difco) agar plates. The typical *Salmonella* colonies on XLD agar are pink with dark black centers and further confirmed by TSI (Difco).

Three to five *Salmonella* isolates per positive sample were selected for serotyping. according to the Kauffmann-White scheme with specific antisera (O) and flagella (H) antisera (S&A reagent Lab Ltd., Bangkok, Thailand) [19].

## Isolation and confirmation of *V. cholerae*

The isolation of *V. cholerae* followed the guidelines of the US FDA BAM with a modification [20]. The homogenized samples of the BPW samples in the previous step were inoculated into 9 ml of alkaline peptone water (APW) (Difco) and incubated at 37˚C for 24 hr. A loopful of APW was streaked onto a thiosulfate-citrate-bile salts-sucrose (TCBS) (Difco) agar plate, and the plate was incubated at 37˚C overnight. The presence of a circular yellow colony on TCBS agar plates is presumptively identified as *V. cholerae*. The isolates were then streaked onto CHROMagar<sup>TM</sup> *Vibrio* (HiMedia Laboratories) for confirmation. After overnight incubation at 37˚C, the colonies of *V. cholerae* appeared in green-blue colonies on CHROMagar<sup>TM</sup> *Vibrio*

agar plates. These colonies were further confirmed with TSI slant agar supplemented with 2% NaCl. The isolates were subsequently purified in TSA agar plates and incubated at 37°C for 24 hr. Within each positive sample, a single colony was stored for further analysis.

## Molecular confirmation of pathogens

All bacterial isolates were re-streaked on nutrient agar (Difco) to ensure purity. Genomic DNA was extracted using a whole-cell boiling method [21]. A single pure colony of cultured isolates was picked and transferred to a 1.5 ml microtube with 120 μl of RNAase-free water and mixed by a vortex. The suspension was boiled for 10 min, immediately placed on ice, and centrifuged at 11,000 g for 5 min. The supernatant was collected as a bacterial DNA template in subsequent analyses.

Molecular confirmation of bacterial isolates was performed by simplex polymerase chain reaction (PCR) method. The biochemically confirmed colonies of *A. hydrophila* were analyzed using genus-specific (*aer*-F/*aer*-R; 5'-CTA CTT TTG CCG GCG AGC GG '3 and 5'-TGA TTC CCG AAG GCA CTC CC '3) and species-specific primers (*ah*-F/*ah*-R; 5'-GAA AGG TTG ATG CCT AAT ACG TA '3 and 5'-CGT GCT GGC AAC AAA GGA CAG '3) [22]. *Salmonella* isolates were tested for the presence of the *invA* gene (*invA*-F/*invA*-R; 5'- GTGAAATTATCGCCACGTTCGGGCAA-'3 and 5'-TCATCGCACCGTCAAAGGAACC-'3) [23]. On the other hand, *V. cholerae* isolates were confirmed using the *ompW* gene (*ompW*-F/*ompW*-R; 5'-CACCAAGAAGGTGACTTTATTGTG-'3 and 5'-GAACTTATA ACCACCCGCG-'3) [24]. The reference stains used in this study were *A. hydrophila* DMST 2798, *Salmonella* Enteritidis DMST 15676, and *V. cholerae* DMST 2873.

## Identification of virulence genes

The simplex PCR technique was performed to determine the presence of virulence genes. The primer sequences used for the detection of virulence genes are previously described in S1 Table [22–28]. Two virulence genes for *A. hydrophila*, two virulence genes, aerolysin (*aero*) and hemolysin (*hly*), were tested with conditions as follows the initial denaturation at 94°C for 3 min, followed by 30 cycles of 94°C for 30 s, 52°C for 30 s, 72°C for 30 s, and a final extension step of 72°C for 10 min.

Three virulence genes of *V. cholerae* were examined, including genes encoded in toxin-co-regulated pilus (*tcpA*), cholera toxin (*ctx*), and hemolysin A (*hlyA*). The PCR condition for *tcpA* and *ctx* was 94°C for 2 min, followed by 30 cycles of 94°C for 60 s, 62°C for 60 s, 72°C for 60 s, and a final extension at 72°C for 10 min [27, 28]. For *hlyA* amplification, the PCR condition was 94°C for 5 min, followed by 35 cycles of 94°C for 60 s, 58°C for 60 s, 72°C for 60 s, and a final extension was carried out at 72°C for 5 min [29].

## Antimicrobial susceptibility test (AST)

Phenotypic characterization of AMR was performed using the agar dilution method according to the Clinical and Laboratory Standards Institute (CLSI) [30]. Twelve antimicrobials tested were selected due to their medical importance for humans, terrestrial, and aquatic animals. Minimum inhibitory concentrations (MIC) were determined. Antimicrobials with (range in μg/ml): ampicillin (AMP; 0.5–1024 μg/ml), chloramphenicol (CHP; 1–256 μg/ml), ciprofloxacin (CIP; 0.015–32 μg/ml), enrofloxacin (ENR; 0.0075–64 μg/ml), florfenicol (FFC; 0.5–512 μg/ml), gentamicin (GEN; 0.25–128 μg/ml), oxolinic acid (OXO; 0.015–128 μg/ml), oxy-tetracycline (OTC; 0.06–512 μg/ml), streptomycin (STR; 1–512 μg/ml), sulfamethoxazole (SMZ; 2–2048 μg/ml), tetracycline (TET; 0.125–512 μg/ml), and trimethoprim (TMP; 0.125–256 μg/ml) were tested. The bacterial susceptibility was determined using CLSI breakpoints

for *Enterobacteriaceae* [30, 31], *V. cholerae* [32], and *A. hydrophila* [33]. *Staphylococcus aureus* ATCC 29213, *Escherichia coli* ATCC 25922, and *Pseudomonas aeruginosa* ATCC 27853 were used as quality control strains.

## ESBL production

The detection of ESBL production involves two steps, which are the screening test and confirmation. For the screening test, the disk diffusion method was used [30]. Three individual cephalosporin disks, including cefotaxime (30 µg) (Oxiod, Hampshire, UK), cefpodoxime (10 µg) (Oxiod), and ceftazidime (30 µg) (Oxoid) were placed on Muller Hilton (MHA) (Difco) agar plates, which were fully spread with a bacterial suspension prepared in 0.9% NaCl solution at 0.5 McFarland standard. The MHA plates were incubated at 37°C for 24 hr. The diameter of the inhibition zone was measured and analyzed according to the CLSI standard. Subsequently, the bacterial isolates, which were resistant to at least one single cephalosporin, were confirmed using the combination disk diffusion method. Two single cephalosporin disks containing cefotaxime (30 µg) (Oxiod) and ceftazidime (30 µg) (Oxiod) were applied in combination with clavulanic acid (10 µg) (Oxiod) to confirm the presence of ESBL production. The isolates were identified as producing ESBL if the difference in diameter of the inhibition zones between the individual cephalosporin disks and the disks with added clavulanic acid was greater than 5 mm.

## Detection of resistance genes and resistance determinants

All primers used in this study are previously described (S2 Table) [34–53]. After, the PCR amplification, the PCR products were separated by gel electrophoresis on a 1.5% (w/v) agarose gel, which was stained with a Redsafe™ nucleic acid staining solution (Intron Biotechnology, Seongnam, Republic of Korea). All DNA fragments were visualized using the Omega Fluor™ gel documentation system. (Aplegen, CA, USA).

## Nucleotide sequencing

*Salmonella* isolates (*n* = 7) were selected to determine mutations in the QRDRs of *gyr*A and *par*C genes. The primers used in this study were previously validated [54]. Seven *Salmonella* isolates were collected from the intestinal tract (*n* = 2), fish meat (*n* = 3), and liver and kidney (*n* = 2). All amplicons were submitted for DNA purification and sequenced (Bionics Co., LTD., Gyeonggi-Do, Republic of Korea). The nucleotide sequences were aligned and compared with references at the National Centre for Biotechnology Information (NCBI) (http://blast.ncbi.nlm.nih.gov/Blast.cgi) using the Molecular Evolutionary Genetic Analysis (MEGA) software version 11. All sequences (accession number OR090930-OR090936) were presented in the NCBI database.

## Statistical analysis

The distribution of bacterial pathogens across different sample types (fish meat, intestine, and liver and kidney), and various sources of tilapia (supermarket VS fresh market) was examined using descriptive statistics. To compare the prevalence of pathogens in fresh markets and supermarkets, the Chi-square test was used and analysed by SPSS software version 22 (IBM Corporation, Armonk, NY, USA). The association between the resistance phenotype and genotype, and virulence genes was determined by logistic regression using Stata version 14.0 (StataCorp, College Station, TX, USA). Logistic regression models were constructed using both forward addition and backward elimination approaches. The significance level of $p < 0.05$ was considered statistical significance.

## Results

### Distribution of *A. hydrophila*, *Salmonella*, and *V. cholerae*

The tilapia collected in this study were of marketable size with an average weight of 741.1 ± 240.6 grams. At the sales point, the fish were mostly displayed on ice (73.2%), while a small proportion of fish were preserved and sold without ice (26.8%). The average ambient air temperature and relative humidity were 28.9 ± 3.3˚C and 63.6 ± 8.9% in the fresh markets and 28.9 ± 1.5˚C and 55.5 ± 7.5% in supermarkets, respectively.

The overall prevalence of *A. hydrophila*, *Salmonella*, and *V. cholerae* were 3.1%, 7.4%, and 8.5%, respectively. These pathogens were significantly higher in samples collected from fresh markets compared to those from supermarkets ($p < 0.05$). And, the prevalence of *Salmonella* ($p = 0.011$) and *V. cholerae* ($p < 0.001$) among fish samples were significantly difference (Fig 1). *Salmonella* serovar Paratyphi B (6.4%), Escanaba (5.7%), Saintpaul (5.7%), Neukoelln (5.0%), and Papuana (5.0%) were commonly observed in this study (S3 Table). All virulence genes detected in these pathogens were 100% positive in *A. hydrophila* (*aero* and *hly*), *Salmonella* (*invA*), and *V. cholerae* (*hlyA* and *ompW*). No detection of epidemic *V. cholerae* carrying *ctx* or *tcpA* were observed.

### AMR of *A. hydrophila*

In this study, the prevalence of ampicillin resistance in *A. hydrophila* was detected as follows: 38.5% in liver and kidney samples, 34.6% in the intestine samples, and 26.9% in the fish meat samples (Fig 1). All isolates were resistant to at least one antimicrobial, with more than half of them classified as MDR (65.4%). The most common resistance was observed in ampicillin (100.0%), and florfenicol (100.0%), with oxytetracycline and tetracycline demonstrating an equivalent resistance rate of 49.9% (Fig 2). Among the antimicrobials tested, none of *A. hydrophila* was resistant to ciprofloxacin, enrofloxacin, gentamicin, and oxolinic acid. Within the eight distinct resistance patterns observed, the majority of *A. hydrophila* were AMP-FFC

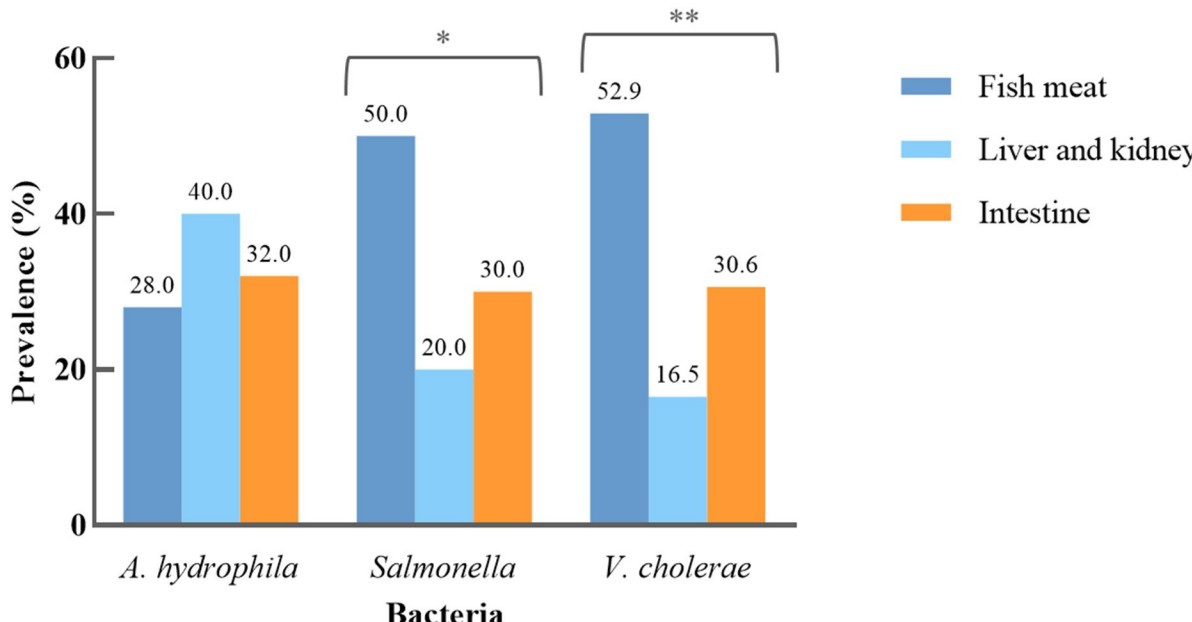

**Fig 1. Prevalence of *A. hydrophila*, *Salmonella*, and *V. cholerae* stratified by fish sample types.**

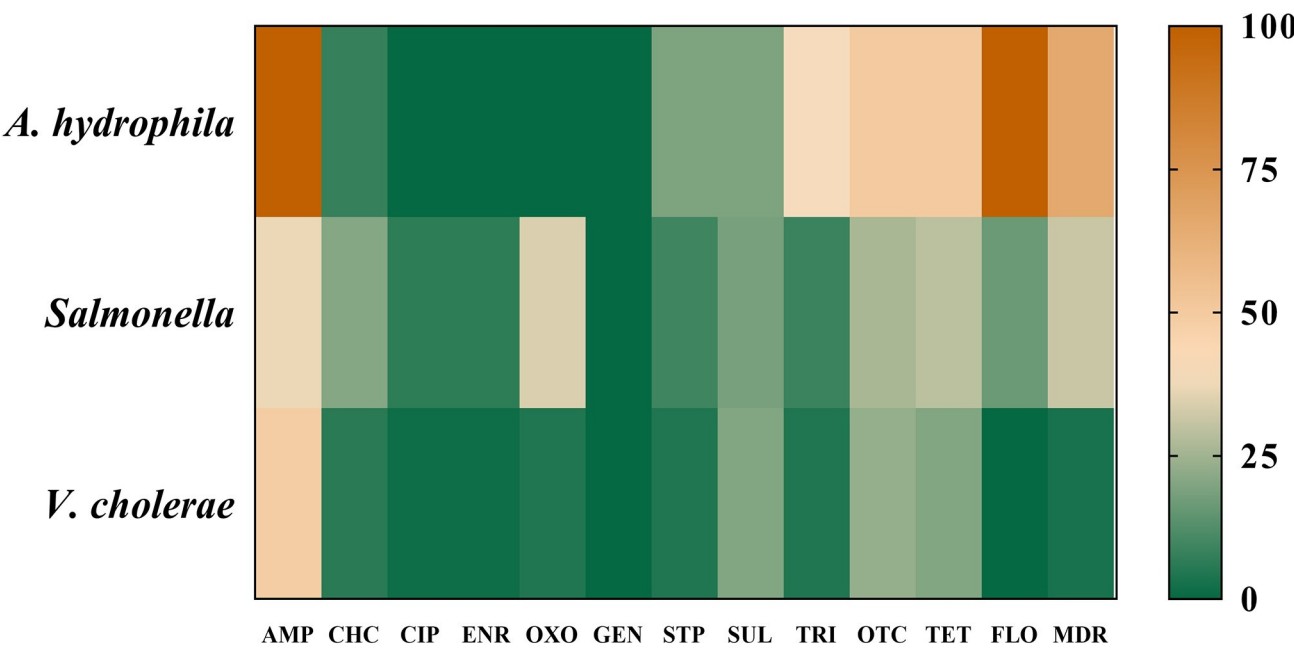

**Fig 2. Resistance phenotypes and MDR of *A. hydrophila* (*n* = 26), *Salmonella* (*n* = 141), and *V. cholerae* (*n* = 70).**

(34.6%) and AMP-FFC-OTC-TET (23.1%) (S4 Table). None of *A. hydrophila* was found to be ESBL producers and carries of any resistance determinants. The most resistant genes among these bacterial isolates were *sul1* (53.8%), *tetA* (50.0%), *sul2* (30.8%), and *florR* (26.9%) (Fig 3).

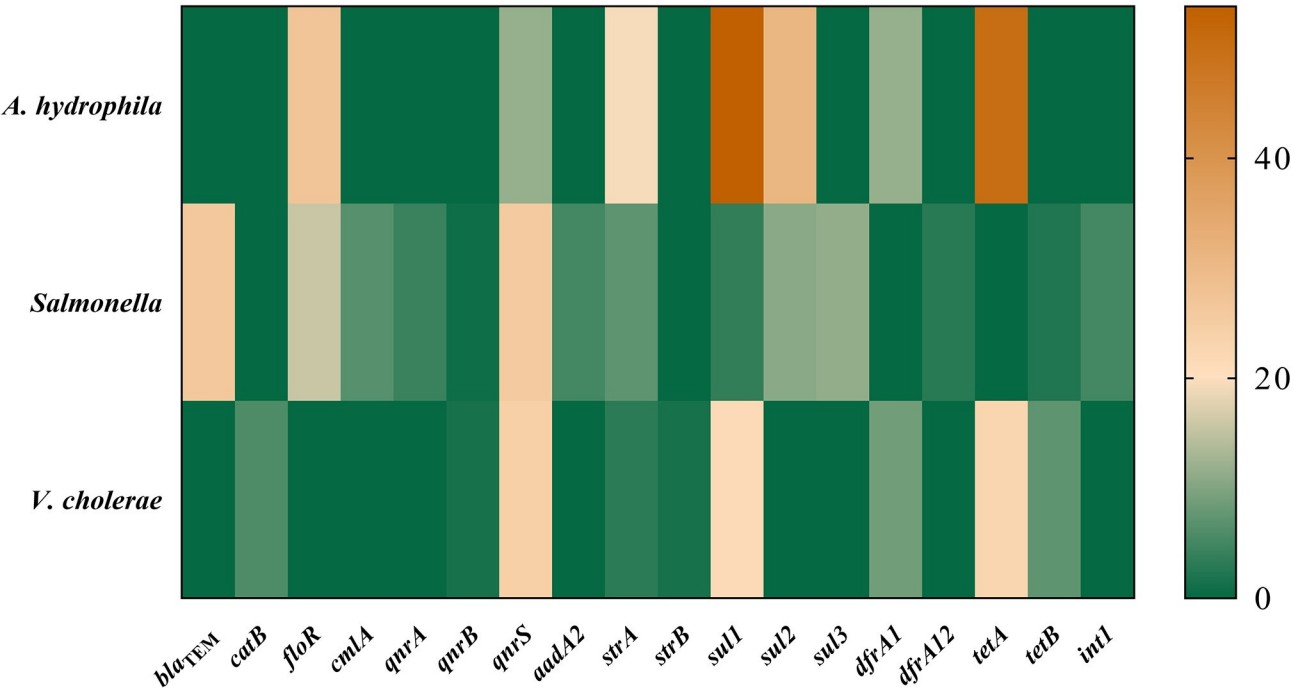

**Fig 3. Resistance genotypes of *A. hydrophila* (*n* = 26), *Salmonella* (*n* = 141), and *V. cholerae* (*n* = 70).**

## AMR of *Salmonella* spp.

All *Salmonella* isolates (*n* = 141) carried the *invA* gene. Almost one-third of *Salmonella* isolates (31.2%) exhibited MDR, and 54.6% of isolates were resistant to at least one antimicrobial (Fig 2). Resistance was frequently identified in isolates against ampicillin (36.8%), oxolinic acid (34.1%), tetracycline (29.1%), and oxytetracycline (26.3%). However, none of the isolates was resistant to gentamicin. Among 32 AMR patterns, the common resistant patterns were OXO (9.9%), and AMP-CHP-FFC-OTC-OXO-TET (7.1%) (S5 Table). None of the *Salmonella* isolates exhibited the ESBL production phenotype. Among the various AMR genotypes, the most predominant resistance genes were $bla_{TEM}$ (26.2%), *qnrS* (25.5%), and *floR* (15.6%). Furthermore, *int1* was detected in 5.0% of *Salmonella* (Fig 3).

## AMR of *V. cholerae*

Of the 70 *V. cholerae* isolates, resistance to ampicillin, oxytetracycline, sulfamethoxazole, and tetracycline was 48.6%, 22.8%, 20.0% and 20.0%, respectively (Fig 2). However, resistance to florfenicol and gentamicin was absent in this collection. Two-thirds of *V. cholerae* isolates (67.1%) were resistant to at least one antimicrobial, while fewer than 6% of the isolates were MDR. Among 13 resistance patterns identified within *V. cholerae*, the most frequent patterns were AMP (25.7%) and AMP-OTC-TET (14.3%) (S6 Table). None of *V. cholerae* was an ESBL producer, and no evidence of resistance determinants was detected within this strain collection. Common resistance genes observed among *V. cholerae* isolates included *qnrS* (24.2%), *tetA* (22.8%), and *sul1* (21.5%) (Fig 3).

## Mutation of QRDRs

Within the *Salmonella* isolates (*n* = 7), comprising four serovar Saintpaul isolates and one isolate each of serovar Derby, Virchow, and Schwabach, the QRDRs, *gyr*A and *par*C, were examined. A single nucleotide mutation within the *gyr*A region was detected in six of seven isolates, specifically at position 248. This mutation involved the substitution of C with A, resulting in an amino acid change from Serine (Ser) to Tyrosine (Tyr) at position 83 with a MIC of between 2–8 μg/ml (Table 1). None of the silent mutations and mutations within the *parC* region were observed in all ciprofloxacin-resistant *Salmonella* isolates.

## Association between phenotypic and genotypic AMR among pathogens

The results of this study revealed that the co-harbour of resistance genes in these ampicillin-resistant pathogens was observed through logistic regression modelling (Table 2). The

**Table 1. Mutation of *gyr*A and *par*C in QRDRs in the ciprofloxacin-resistant *Salmonella* isolates (*n* = 7).**

| Source | Type of sample (*n*) | MIC[a] (μg/ml) | Serotype (*n*) | Nucleotide alteration | |
|---|---|---|---|---|---|
| | | | | *gyr*A | *par*C |
| Fresh market | Meat (1) | 2 | Virchow (1) | ND | ND[b] |
| Fresh market | Meat (1) Intestine (1) Liver and kidney (1) | 2 2 2 | Saintpaul (1) Derby (1) Schwabach (1) | C-248-A | ND[b] |
| Fresh market | Meat (1) Liver and kidney (1) | 4 | Saintpaul (2) | C-248-A | ND[b] |
| Supermarket | Intestine (1) | 8 | Saintpaul (1) | C-248-A | ND[b] |

[a]MIC: Minimum inhibitory concentration

[b]ND: non-detected

**Table 2. The association between phenotypic ampicillin resistance and genotypic resistance classified by bacterial pathogens using logistic regression analysis (*n* = 237).**

| Predictor | Odds ratio | Std. Err.[a] | 95% C.I.[b] | *p*-value |
|:---:|:---:|:---:|:---:|:---:|
| *bla*$_{\text{TEM}}$ | 156.68 | 77.98 | 59.07 to 415.61 | <0.0001 |
| *tetA* | 7.10 | 4.08 | 2.31 to 21.89 | 0.001 |
| *sul1* | 4.59 | 0.43 | 3.81 to 5.53 | <0.0001 |
| *sul2* | 6.16 | 0.68 | 4.96 to 7.65 | <0.0001 |
| *sul3* | 157.72 | 94.96 | 48.46 to 513.34 | <0.0001 |
| Constant | 0.08 | 0.02 | 0.05 to 0.12 | <0.0001 |

AIC[c] = 158.71

[a]Std. Err.: Standard error

[b]CI: confidence interval

[c]AIC: Akaike's Information Criteria

presence of ampicillin-resistant bacteria was positively associated with the existence of specific genes such as *bla*$_{\text{TEM}}$ (*p* < 0.0001, OR = 156.68), *tetA* (*p* = 0.001, OR = 7.10), *sul1* (*p* < 0.0001, OR = 4.59), *sul2* (*p* < 0.0001, OR = 6.16), and *sul3* (*p* < 0.0001, OR = 157.72) compared to ampicillin-susceptible pathogens.

## Discussion

In this study, the overall prevalence of *A. hydrophila*, *Salmonella*, and *V. cholerae* was less than 10%, which was lower than a previous report in cage-cultured tilapia in Thailand [55]. Moreover, observed detection of these pathogens was lower than the tilapia sold in markets in Brazil and Malaysia [9, 10]. The occurrence of these three pathogens was significantly higher in samples purchased from the fresh market compared to those in supermarkets. These findings contrasted with a study of *A. hydrophila*, *Salmonella*, and *Vibrio* spp. in uncooked seafood sold in Bangkok, which indicated a similar level of this bacterial contamination between fresh markets and supermarkets [56]. This variation could indicate increased awareness of hygiene practices during fish preparation and storage processes in tilapia sold in supermarkets.

*A. hydrophila* is a common pathogen in both fish and humans circulating on tilapia farms. Although the abundance of this bacterium in fish and aquatic systems has been previously reported, the prevalence of *A. hydrophila* observed in this study remains notably low. This finding was in contrast to a study of prevalent *Aeromonas* spp. in healthy tilapia in Tanzania [57]. The prevalence of *Aeromonads* can vary due to the disease status of the farm and the geographical distribution. In this study, *A. hydrophila* mainly carried two virulence genes, including *aer* and *hly*, which was similar to the finding in tilapia farms in Tanzania [57]. Infection with these virulent *Aeromonas* can cause Aeromonads enteritis in fish consumers.

All isolates of *A. hydrophila* examined in this study showed resistance to ampicillin and florfenicol. The high resistance to beta-lactamase and phenicols corresponded to previous studies in Egypt [4, 58]. This implied that these antimicrobials may currently be ineffective for treating infection. This study showed that all *A. hydrophila* were susceptible to two fluoroquinolones, including ciprofloxacin and enrofloxacin, which were commonly used for the treatment of *A. hydrophila* in aquaculture [4]. Consequently, these two antibiotics are currently effective in treating *A. hydrophila* infection in tilapia farms. According to the CLSI guideline, *A. hydrophila* is the only bacteria recommended for AMR monitoring in aquatic animals [33]. However, tilapia can be naturally infected with other *Aeromonas* spp., especially *A. veronii* and *A. jandaei*, which are prevalent fishborne pathogens and have been reported in tilapia farms throughout Thailand [59]. More than

half of the isolates of *A. hydrophila* in this study (65.4%) showed MDR. A previous study observed that *Aeromonas* spp. can acquire new resistance genes from other bacteria, especially *Enterobacteriaceae*, through transferable plasmids [60]. This would be a driving factor contributing to the widespread spread of MDR *A. hydrophila* [58, 61]. Therefore, *A. hydrophila* can serve as a reservoir for the resistance gene pool within the freshwater aquatic system.

*Salmonella* is responsible for a primary cause of bacterial diarrhea in humans worldwide. In this study, a high prevalence of *Salmonella* (7.4%) was detected in tilapia. Although this prevalence was lower than in a previous study of domestic (19.4%) and imported tilapia (33.3%) sold in the United States, respectively [62]. The most common serovars of *Salmonella* observed in this study, including Paratyphi B, Escanaba, and Saintpaul, have previously been reported in aquatic products and tilapia farms in Thailand [55]. Importantly, *S.* Paratyphi B has been implicated in numerous foodborne outbreaks due to fish consumption [63]. The contamination of the production system by *Salmonella* can occur during both the pre-harvest and harvesting stages. To mitigate the potential for *Salmonella* infections in humans, it is recommended to trace the source of contamination.

This study observed a high prevalence of *Salmonella* resistance to ampicillin, oxolinic acid, tetracycline, and oxytetracycline. High resistance to ampicillin would result from intrinsic resistance to *Salmonella*. In addition, 34.1% of *Salmonella* in this study exhibited resistance to oxolinic acid, which belongs to the first-generation of quinolone. On the contrary, they were more susceptible to second-generation fluoroquinolones, including ciprofloxacin and enrofloxacin. The explanation for this inconsistency was that the modified structure of the second-generation fluoroquinolone resulted in a more potent and broader efficacy against Gram-negative bacteria compared to oxolinic acid [64]. The high resistance of *Salmonella* to tetracycline analogues observed in this study was consistent with the observed resistance in retail aquatic products in Nigeria [65]. Furthermore, resistance to these antimicrobials has been identified the prevailing trend among bacteria isolated from retail meat [14, 65]. In this study, *Salmonella* isolates showed three frequent resistance genes, including $bla_{TEM}$, *qnrS*, and *floR*. The presence of the first two resistance genes corresponded to the observed phenotypic resistance and was consistent with the most common genes detected in aquatic products [66]. Florfenicol is currently used in Thai swine farms and consequently enters the environment through manure. The circulation of this plasmid-borne resistance reflected the continued spread of these genes from livestock to the environment and potentially into aquaculture, which may have an additional risk of AMR infections through the consumption of aquatic products. The results obtained from this study revealed that some isolates carrying *int1* were susceptible to AMR and the development of MDR. On the contrary, the abundance of *Salmonella* carrying *int1* has been reported in freshwater fish in South Africa and France [67, 68]. None of these isolates was an ESBL producer, and all isolates were negative for colistin resistance genes. However, *mcr-1* and *mcr-3* were widely reported in fish in Thailand and Vietnam, addressing fish could represent a novel source of clinically important resistant bacteria [13, 69, 70].

The prevalence of non-O1/non-O139 *V. cholerae* observed in this study was 8.5%, which was similar to a report in China [71]. All isolates of *V. cholerae* lacked the *ctx* and *tcpA* genes, indicating that there were non-cholera *Vibrios* in these fish. These *Vibrio* strains are commonly abundant in freshwater and coastal aquatic system but are occasionally observed in clinical cases [11]. Some isolates of *V. cholerae* harbored *hlyA*, which are associated with watery diarrhea and sporadic outbreaks observed in Thailand [72]. Almost half of the *V. cholerae* isolated from tilapia was resistant to ampicillin, in contrast to previous studies that showed a high susceptibility of *V. cholerae* isolated from fish and coastal waters [11, 73]. Furthermore, a considerably high percentage of isolates of *V. cholerae* showed resistance to oxytetracycline, sulfamethoxazole, and tetracycline. Although less than 3% of *V. cholerae* isolates were MDR strains,

it should be noted that tilapia may be at high risk for AMR *V. cholerae*. The common resistance genes identified in this study were *qnrS*, *tetA*, and *sul1*, which was consistent with a previous study in tilapia in Egypt [74]. The presence of *tet* genes within *V. cholerae*, which conferred resistance to tetracyclines was in agreement with the concern of resistant-*V. cholerae* in clinical isolates worldwide [75].

One-fifth of *V. cholerae* harbored *sul1*, which was consistent with a study of *Vibrio* spp. isolated from fish in South Africa [10]. *Vibrios* are well known for their ability to carry resistance determinants by integrons and the SXT element. The SXT element was recovered from clinical *V. cholerae* in Thailand; however, none of the mobile genetic elements were detected in *V. cholerae* in this study [76]. The highest phenotypic resistance observed among *V. cholerae* isolates in this study was beta-lactamase resistance, despite the absence of $bla_{TEM}$ detection. The resistant discrimination of this observation may be mediated by the carrying of the cassette-encoded β-lactamases (CARB) gene, which belongs to a class A beta-lactamase that shares similarity with $bla_{TEM}$, such as $bla_{CARB-7}$ and $bla_{CARB-9}$. These $bla_{CARB}$ genes have been regularly reported in clinical and environmental *V. cholerae* in Australia [77]. Many cases of cholera are often associated with aquatic environments, such as the consumption of fish products or exposure to recreational waters [9]. Therefore, the relationship between clinical strains and environmental *V. cholerae* isolates should be elaborated to understand their dynamic shift and the plausible role they play as reservoirs of virulence and resistance factors.

The determination of amino acid changes in the QRDRs was mandatory to capture trends and identify novel mutations conferred to different levels of resistance levels. In this study, only *gyr*A mutations were observed at position 83 (Ser to Ile), similar findings in *Salmonella* isolates obtained from pigs, chickens, and humans in Thailand [14]. The coexistence of resistance genes to multiple antimicrobial classes, including beta-lactams, tetracyclines, and sulfonamides, was confirmed by logistic regression analysis. Ampicillin resistance has been reported to be the dominant resistance phenotype in tilapia in *Salmonella* in Brazil and *A. hydrophila* in Egypt [7, 58]. The predominant resistance genes, $bla_{TEM}$, *sul1*, and *tetA*, were commonly reported in fish populations worldwide. For example, *sul* genes were the most abundant resistance genes in non-cholera *Vibrio* isolates from fish in South Africa, while $bla_{TEM}$ and *tet* genes were mainly presented in *Salmonella* in tilapia farms in Thailand [10, 13]. In addition, the co-occurrence of $bla_{TEM}$ and *tetA* has been previously observed in *E. coli* isolated from seafood in Nigeria, suggesting a trend of co-resistance [78]. The investigation of mechanisms contributing to this observation, such as the role of relevant plasmids or conjugative elements, would be valuable in mitigating the development of AMR in fish and aquatic products. Therefore, it is important to emphasize the proper storage practice and mitigation of cross-contamination in fish handlers to ensure the safety of fish as a consumable food source. Consumers should also be educated on the significance of consuming fully cooked fish to mitigate the potential risks of fish-related foodborne illnesses. A comprehensive monitoring program for AMR in the retail sales of aquatic products should be implemented on a national scale and mandated for all individuals engaged in the trade of aquatic products. To effectively address AMR challenges in fish, it is important to raise awareness among fishmongers and farmers regarding pathogens and the associated AMR risks. Furthermore, the active participation of various stakeholders is crucial for identifying gaps related to AMR and providing policymakers with essential insights to tackle AMR issues in the domain of fish consumption.

## Conclusions

This study signified a high prevalence of AMR pathogen contamination in Nile tilapia available in markets and supermarkets in Thailand. To reduce the risk of AMR transmission and ensure

the safety of tilapia consumption, subsequent implementations must be rigorously initiated. In retail markets, it is essential to enhance the overall hygiene of fish stalls, storage, and transportation, maintaining a consistent cooling temperature. Furthermore, fish advisories should aim to educate consumers about the proper evisceration of fish, thorough cleaning of fish skin and abdominal cavity, cooking at the appropriate time and temperature, and preventing cross-contamination during the cooking process. The results of this study can be useful in the accomplishment of Thailand's National Action Plan (NAP) under the One Health framework. To proactively address AMR in fish, it is imperative to prioritize the specification of target resistance in each bacterial species as a warning sign for emerging AMR.

## Supporting information

**S1 Table. Primers used for virulence gene detection of *A. hydrophila*, *Salmonella* spp., and *V. cholerae*.**
(DOCX)

**S2 Table. Primers used for detection of AMR genes of *A. hydrophila*, *Salmonella* spp., and *V. cholerae*.**
(DOCX)

**S3 Table. *Salmonella* serovars isolated from Nile tilapia (*n* = 141).**
(DOCX)

**S4 Table. AMR patterns of *A. hydrophila* isolated from Nile tilapia (*n* = 26).**
(DOCX)

**S5 Table. AMR patterns of *Salmonella* isolated from Nile tilapia (*n* = 141).**
(DOCX)

**S6 Table. AMR patterns of *V. cholerae* isolated from Nile tilapia (*n* = 70).**
(DOCX)

## Acknowledgments

We thank Sutida Chalee and Saran Anantavirun for sample collection and laboratory assistance.

## Author Contributions

**Conceptualization:** Saharuetai Jeamsripong.

**Data curation:** Jarukorn Sripradite, Varangkana Thaotumpitak, Saharuetai Jeamsripong.

**Formal analysis:** Varangkana Thaotumpitak, Saharuetai Jeamsripong.

**Funding acquisition:** Jarukorn Sripradite, Saharuetai Jeamsripong.

**Investigation:** Jarukorn Sripradite, Varangkana Thaotumpitak, Saharuetai Jeamsripong.

**Methodology:** Jarukorn Sripradite, Woranich Hinthong.

**Resources:** Edward R. Atwill, Saharuetai Jeamsripong.

**Supervision:** Saharuetai Jeamsripong.

**Validation:** Saharuetai Jeamsripong.

**Visualization:** Saharuetai Jeamsripong.

**Writing – original draft:** Jarukorn Sripradite, Varangkana Thaotumpitak, Saharuetai
    Jeamsripong.

**Writing – review & editing:** Edward R. Atwill, Saharuetai Jeamsripong.

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
