## [Decision Letter · Decision Letter 0]

13 Oct 2023

PONE-D-23-29694Bacterial pathogens and antimicrobial resistance in retail Nile tilapia (Oreochromis spp.) as potential sources of foodborne illnessPLOS ONE

Dear Dr. Jeamsripong,

Thank you for submitting your manuscript to PLOS ONE. After careful consideration, we feel that it has merit but does not fully meet PLOS ONE’s publication criteria as it currently stands. Therefore, we invite you to submit a revised version of the manuscript that addresses the points raised during the review process.

Reviewers have identified number of gaps and raised many questions. Please address all reviewer comments.

We look forward to receiving your revised manuscript.

Kind regards,

Iddya Karunasagar

Academic Editor

PLOS ONE

Journal Requirements:

   "This study was supported by Thailand Science Research and Innovation Fund Chulalongkorn University (HEA663100108), National Research Council of Thailand (NTCT) (N42A660897), University of California Davis (A22-3754-S001), and Rachadapisek Sompote Fund Chulalongkorn University (GR_62_37_31_02) and College of Industrial Technology, King Mongkut’s University of Technology North Bangkok (Res-CIT0232/2019). We thank Sutida Chalee and Saran Anantavirun for sample collection and laboratory assistance."

Additional Editor Comments:

The reviewers have raised very important questions. Please provide information on the positive controls used and justify the choice of virulence genes. Please provide information on the farm source of samples. Please address all reviewer comments point by point.

Reviewers' comments:

Reviewer's Responses to Questions

**Comments to the Author**

1. Is the manuscript technically sound, and do the data support the conclusions?

Reviewer #1: Partly

Reviewer #2: Yes

2. Has the statistical analysis been performed appropriately and rigorously? 

Reviewer #1: Yes

Reviewer #2: Yes

3. Have the authors made all data underlying the findings in their manuscript fully available?

Reviewer #1: Yes

Reviewer #2: Yes

4. Is the manuscript presented in an intelligible fashion and written in standard English?

Reviewer #1: Yes

Reviewer #2: Yes

5. Review Comments to the Author

Reviewer #1: �Overall writing of the manuscript is good

Several similar reports have been published in recent years, hence the work has limited novelty

Focusing more on tilapia aquaculture without considering the farm source of the samples

The overall prevalence of targeted bacterial species was low compared to previous reports. Authors could have easily highlighted the safety of the fish samples comparative to fish from other markets.

The study observed no cholera causing vibrios among the isolates, which was not highlighted

It is good that the study concluded suggesting better cooking methods to avoid possible food borne infection.

It looks like the authors have ended the discussion abruptly.

The conclusion part of the manuscript not corroborating with the results of the study

As indicated by the authors the results of the study might be helpful to the Thailand’s National Action Plan of One Health Framework but dose-not advance the knowledge to the general reader.

Reviewer #2: 1. The authors would have mentioned the details of positive controls used for the study.

2. What is the rationale behind the selection of three pathogens? The authors should emphasize this in the manuscript.

3. Why have authors restricted to only a few virulence genes in 3 pathogens? Especially in Salmonella and Aeromonas hydrophillaLine 198 rnase should be written as RNAase

4. Few typographical errors in the manuscript need to be taken care

5. It is better if the Authors use h or hr or hours in the manuscript

6. PLOS authors have the option to publish the peer review history of their article (what does this mean?). If published, this will include your full peer review and any attached files.

Reviewer #1: No

Reviewer #2: No

---

## [Author Response · Author response to Decision Letter 0]

14 Nov 2023

Manuscript ID: PONE-D-23-29694: Bacterial pathogens and antimicrobial resistance in retail Nile tilapia (Oreochromis spp.) as potential sources of foodborne illness

Review comments to the author 

Reviewer#1

1. Overall writing of the manuscript is good. 

Thank you for your reviewing.

2. Several similar reports have been published in recent years, hence the work has limited novelty

This study offered an original contribution by demonstrating the safety of tilapia as a suitable food source for human consumption and establishing the extent of microbial contamination in tilapia.

3. Focusing more on tilapia aquaculture without considering the farm source of the samples

While this study did not specifically address farms, it is worth noting that different farms in the study exhibited similar farm management systems and biosecurity measures. Therefore, further investigation into diverse farm management practices should be investigated.

4. The overall prevalence of targeted bacterial species was low compared to previous reports. Authors could have easily highlighted the safety of the fish samples comparative to fish from other markets.

The safety of consuming fish in Thailand was found to be comparable with safety levels observed in other countries and indicated in lines 389-390.

5. The study observed no cholera causing vibrios among the isolates, which was not highlighted

The sentences concerning the absence of cholera-causing vibrios were stated, "All isolates of V. cholerae lacked the ctx and tcpA genes, indicating the presence of non-cholera vibrios in these fish" indicating in line 462-463.

6. It is good that the study concluded suggesting better cooking methods to avoid possible food borne infection.

Thank you for your comment.

7. It looks like the authors have ended the discussion abruptly.

We have provided the discussion sentences to be added in lines 512-518. Here they are incorporated into the text: “A comprehensive monitoring program for AMR in the retail sales of aquatic products should be implemented on a national scale and mandated for all individuals engaged in the trade of aquatic products. To effectively address AMR challenges in fish, it is important to raise awareness among fishmongers and farmers regarding pathogens and the associated AMR risks. Furthermore, the active participation of various stakeholders is crucial for identifying gaps related to AMR and providing policymakers with essential insights to tackle AMR issues in the domain of fish consumption.”

8. The conclusion part of the manuscript not corroborating with the results of the study

The conclusion has been modified to align with the findings of this study.

9. As indicated by the authors the results of the study might be helpful to the Thailand’s National Action Plan of One Health Framework but dose-not advance the knowledge to the general reader.

In lines 524-528, these sentences have been adjusted in the conclusion to target general consumer education: “In retail markets, it is essential to enhance the overall hygiene of fish stalls, storage, and transportation, maintaining a consistent cooling temperature. Furthermore, fish advisories should aim to educate consumers about the proper evisceration of fish, thorough cleaning of fish skin and abdominal cavity, cooking at the appropriate time and temperature, and preventing cross-contamination during the cooking process.”

10. However, the study has required experimental design and the results have been analysed appropriately.

Thanks for your comment.

11. Since the observed prevalence of the pathogenic bacteria are low in the study, change in the title may be considered.

The title of this manuscript is revised as “Distribution of bacteria and antimicrobial resistance in retail Nile tilapia (Oreochromis spp.) as potential sources of foodborne illness”.

12. After carefully addressing the suggestions the manuscript may be considered for publication in the any of the regional journals 

Thanks for your comment.

Specific comments

Introduction

1. Line no. 24-26: First two sentences are very generic in nature may be deleted from the abstract

The initial two sentences have been removed.

2. Line no. 52-55: The bacteria isolated in the study and the identified AMR in the present study are not traced to farming system/production facilities. Hence, starting the manuscript with aquaculture intensification may not be appropriate. Instead highlights may be provided regarding the post harvest handling of fish and fishery products. Samples in the study are taken from markets and results suggest significant difference between the samples from fresh and supermarkets. 

The importance of post-harvest fish handling has been added in line 54-59as follows: “Fish consumption has been increased due to its perceived health benefits and affordability. However, it is important to note that microbial contaminants in fish have been identified as sources of foodborne outbreaks on a global scale. Additionally, the emergence of antimicrobial resistance (AMR) in aquatic products poses a novel human health risk, with improper storage and handling practices between fresh and supermarkets playing a significant role in this concern.” 

Materials and methods

- Details of bacterial isolations are given in too details, unless any modifications in the standard protocols are made in the study, it is advisable to provide the references only.

The isolation and serotyping methods of Salmonella have been concise.

Results 

1. Line no. 285: fish were displayed without ice???

The sentence "fish were displayed without ice" has been revised to "fish were preserved and sold without ice (26.8%).

2. Table 1 & 2: Since the primers are published earlier, mentioning the references may be sufficient.

To ensure clarity, we have included all primer details and references in Supplementary Tables (S1 and S2), relocating them from Tables 1 and 2.

3. Table 4: This may be provided as supplementary information only, not in the main text.

Table 4, which contains information on Salmonella serovars, has been relocated to S3 Table in the supplementary information.

4. Fig 1 &2: Classifying the prevalence of AMR and the genes based on the tissue origin of the isolates may not be appropriate.

All the figures in this study have been newly created. The first figure illustrates the prevalence of these three pathogens in tilapia, while Fig 2 and 3 show the resistance phenotype and genotype of the pathogens, respectively. 

Discussion

1. Line No. 395-399: Since the farm source of the samples used in the study are not available, assuming that the AMR bacteria were due to unscientific use of antibiotics in the culture system may not be appropriate.

The previous sentences have been removed, and the following sentences have been added to lines 437-444 for clarification: "The high resistance of Salmonella to tetracycline analogues observed in this study was consistency with the observed resistance in retail aquatic products in Nigeria [64]. Furthermore, resistance to these antimicrobials has been identified as a prevailing trend among bacteria isolated from retail meat [14, 64].”

2. Line No. 403-405: Correlating the presence of resistance gene against particular antibiotic in an isolate to its use in the farming system without actual information on the farm usage data may not be appropriate. 

The sentence providing information about antibiotic usage in farms has been removed.

3. Line No. 459-461: Since not data related to farming activity was collected in the study, it may not be appropriate to suggest the observations are due to usage of antibiotics in culture operations.

The sentence indicating the usage of antibiotics in culture operations has been removed, and the following sentence has been added to lines 502-508: "The investigation of mechanisms contributing to this observation, such as the role of relevant plasmids or conjugative elements, would be valuable in mitigating the development of AMR in fish and aquatic products."

4. Line No. 459-464: first sentence indicate frequent use of antibiotics in farms while the next sentence suggest need for better handling practices to reduce the cross contamination. These two sentences are unrelated.

The first sentence has been deleted, and the connecting sentence mentioned earlier in comment No. 3 has been added. 

References

- Total number of references cited by be reduced substantially.

The number of references is reduced as suggested.

Reviewer#2

1. The authors would have mentioned the details of positive controls used for the study.

The three positive control strains used in detection of bacteria and antimicrobial susceptibility test were mentioned in lines 251-252.

2. What is the rationale behind the selection of three pathogens? The authors should emphasize this in the manuscript.

The rationale for selecting these pathogens has been identified in the following lines 68-70 and the below sentence was added for clarification in lines 138-140. “In a previous investigation on tilapia farming in Thailand, three pathogens showed unique profiles of AMR. However, this information has not been sufficiently recorded in tilapia available for human consumption in retail.”

3. Why have authors restricted to only a few virulence genes in 3 pathogens? Especially in Salmonella and Aeromonas hydrophilla

The study focused on a limited number of virulence genes, as the chosen virulence genes are prevalent in bacteria isolated from Nile tilapia and have connections to clinical symptoms in human cases. This explanation has been added in lines 136-138: "Virulence genes (A. hydrophila: aero and hly; Salmonella: invA; V. cholerae: tcpA, ctx, and hlyA), were chosen based on their prevalence and linked to the pathogenesis of diseases in humans.”Ss

4. Line 198 rnase should be written as RNAase

The errors have been corrected.

5. Few typographical errors in the manuscript need to be taken care

The authors recheck all typographical errors throughout this manuscript.

6. It is better if the Authors use h or hr or hours in the manuscript

The abbreviation "hr" has been consistently used in place of "hours" throughout this manuscript.

---

## [Decision Letter · Decision Letter 1]

14 Dec 2023

PONE-D-23-29694R1Distribution of bacteria and antimicrobial resistance in retail Nile tilapia (Oreochromis spp.) as potential sources of foodborne illnessPLOS ONE

Dear Dr. Jeamsripong,

Thank you for submitting your manuscript to PLOS ONE. After careful consideration, we feel that it has merit but does not fully meet PLOS ONE’s publication criteria as it currently stands. Therefore, we invite you to submit a revised version of the manuscript that addresses the points raised during the review process.

Please highlight the novelty of this study vs other published studies in the area. Please also address other comments of the reviewer.  

We look forward to receiving your revised manuscript.

Kind regards,

Iddya Karunasagar

Academic Editor

PLOS ONE

**Additional Editor Comments:**

Please see reviewer comments. Please highlight the novelty of this study vs the article mentioned by the reviewer. Please address all comments of the reviewer.

Reviewers' comments:

Reviewer's Responses to Questions

**Comments to the Author**

1. If the authors have adequately addressed your comments raised in a previous round of review and you feel that this manuscript is now acceptable for publication, you may indicate that here to bypass the “Comments to the Author” section, enter your conflict of interest statement in the “Confidential to Editor” section, and submit your "Accept" recommendation.

Reviewer #2: (No Response)

2. Is the manuscript technically sound, and do the data support the conclusions?

Reviewer #2: Yes

3. Has the statistical analysis been performed appropriately and rigorously? 

Reviewer #2: Yes

4. Have the authors made all data underlying the findings in their manuscript fully available?

Reviewer #2: Yes

5. Is the manuscript presented in an intelligible fashion and written in standard English?

Reviewer #2: Yes

6. Review Comments to the Author

Reviewer #2: Did the standard cultures of A. hydrophila, V. cholerae and Salmonella used in the study? if so please specify

While writing the name of the genes, do not italicize the subunit only the name of the gene should be italic for ex. gyr A where gyr need to be italic not A

highlight the novelity in the introduction section because we can find similar kind of work in the database

https://doi.org/10.3389/frabi.2023.1156258

10.7717/peerj.14896

7. PLOS authors have the option to publish the peer review history of their article (what does this mean?). If published, this will include your full peer review and any attached files.

Reviewer #2: No

---

## [Author Response · Author response to Decision Letter 1]

25 Jan 2024

1. Did the standard cultures of A. hydrophila, V. cholerae and Salmonella used in the study? if so please specify. 

 The standard cultures of A. hydrophila, V. cholerae, and Salmonella were indicated in materials and methods section (Line 212-213). “The reference stains used in this study were A. hydrophila DMST 2798, Salmonella Enteritidis DMST 15676, and V. cholerae DMST 2873.”

2. While writing the name of the genes, do not italicize the subunit only the name of the gene should be italic for ex. gyr A where gyr need to be italic not A 

 The errors have been corrected throughout manuscript included Table2.

3. Highlight the novelity in the introduction section because we can find similar kind of work in the database

https://doi.org/10.3389/frabi.2023.1156258

10.7717/peerj.14896 

 The novelty of this study is indicated in Line 130-135.

“Few studies have conducted investigations into AMR pathogens among retail tilapia intended for human consumption, and the risk of AMR transmission from tilapia consumption has been insufficiently documented in Thailand.”

Editor 

1. Please highlight the novelty of this study vs other published studies in the area. 

Please also address other comments of the reviewer. 

 The sentence relating to the novelty of this study is modified in Line 130-135.

“Few studies have conducted investigations into AMR pathogens among retail tilapia intended for human consumption, and the risk of AMR transmission from tilapia consumption has been insufficiently documented in Thailand.”

2. If applicable, we recommend that you deposit your laboratory protocols in protocols.io to enhance the reproducibility of your results.

 All bacterial determination and confirmation, antimicrobial susceptibility testing, and detection of AMR and virulence genes in this study followed standard methods, which are available in the manuscript. However, specific laboratory protocols are available upon request.

---

## [Decision Letter · Decision Letter 2]

20 Feb 2024

Distribution of bacteria and antimicrobial resistance in retail Nile tilapia (Oreochromis spp.) as potential sources of foodborne illness

PONE-D-23-29694R2

Dear Dr. Jeamsripong,

We’re pleased to inform you that your manuscript has been judged scientifically suitable for publication and will be formally accepted for publication once it meets all outstanding technical requirements.

Kind regards,

Iddya Karunasagar

Academic Editor

PLOS ONE

Additional Editor Comments (optional):

All reviewer comments have been addressed.

Reviewers' comments:

Reviewer's Responses to Questions

**Comments to the Author**

1. If the authors have adequately addressed your comments raised in a previous round of review and you feel that this manuscript is now acceptable for publication, you may indicate that here to bypass the “Comments to the Author” section, enter your conflict of interest statement in the “Confidential to Editor” section, and submit your "Accept" recommendation.

Reviewer #2: All comments have been addressed

2. Is the manuscript technically sound, and do the data support the conclusions?

Reviewer #2: Yes

3. Has the statistical analysis been performed appropriately and rigorously? 

Reviewer #2: Yes

4. Have the authors made all data underlying the findings in their manuscript fully available?

Reviewer #2: Yes

5. Is the manuscript presented in an intelligible fashion and written in standard English?

Reviewer #2: Yes

6. Review Comments to the Author

Reviewer #2: (No Response)

7. PLOS authors have the option to publish the peer review history of their article (what does this mean?). If published, this will include your full peer review and any attached files.

Reviewer #2: No

---

## [Editor Report · Acceptance letter]

22 Mar 2024

PONE-D-23-29694R2 

PLOS ONE

Dear Dr. Jeamsripong, 

I'm pleased to inform you that your manuscript has been deemed suitable for publication in PLOS ONE. Congratulations! Your manuscript is now being handed over to our production team.

Kind regards, 

on behalf of

Dr. Iddya Karunasagar 

Academic Editor

PLOS ONE